# Plant Species as Potential Forage for Honey Bees in the Al-Baha Mountain Region in Southwestern Saudi Arabia

**DOI:** 10.3390/plants12061402

**Published:** 2023-03-22

**Authors:** Ahmad A. Al-Ghamdi, Nageeb A. Al-Sagheer

**Affiliations:** 1College of Food and Agriculture Sciences, King Saud University, Riyadh 11451, Saudi Arabia; aalkhazim1@gmail.com; 2Biology Department, Faculty of Science and Arts in Qilwah, Albaha University (BU), Qilwah 65565, Saudi Arabia; 3Agricultural Research and Extension Authority (AREA), Dhamar 87148, Yemen

**Keywords:** honey production, nectar, pollen, plant checklist, propolis

## Abstract

The contribution of bee forages in the form of nectar, pollen, and propolis to beekeeping development depends on plant species diversity. The data concerning the increase in honey production in southwestern Saudi Arabia, which was unexpected with the deterioration of the vegetation cover, becomes a concrete background for this study, which planned to list the bee plant species contributing as sources of nectar, pollen, and propolis. The sampling method followed a purposive random sampling approach, and 20 × 20 m plots were considered with a total of 450 sample plots. Bee forage plants were identified based on flower morphology and honey bees’ actions during floral visits at active foraging hours. A checklist of bee forages containing 268 plants species belonging to 62 families was documented. The number of pollen source plants (122) was more than nectar (92) and propolis (10) source plants. Regarding seasonal distribution, spring and winter were relatively good seasons for honey bees in terms of pollen, nectar, and propolis availability. Generally, this study is an essential step towards understanding, conserving, and rehabilitating plant species providing nectar, forage, and propolis to honey bees in Al-Baha Region of Saudi Arabia.

## 1. Introduction

Al-Baha is one of the regions in the Kingdom of Saudi Arabia characterized by arid and semi-arid climatic types. This region has relatively great potential for beekeeping with its rich vegetation and diverse environmental conditions. Vegetation diversity is regarded as one of the most important comparative features distinguishing this region from others; wild plants can be found in the deserts, villages, oases, plains, valleys, and mountains of Al-Baha’s landscapes [1]. As a result of its rich floral resource, beekeeping in Al-Baha is one of the most popular and socially accepted livelihood activities in both the Sarawat mountainous and the Tehama coastal areas, which constitute the main apicultural landscapes in the region [2]. A recent study conducted by Al-Ghamdi et al. [3] also reported the existence of enormous honey bee floral resources in the region. However, most of the plant habitats are vulnerable to deterioration and vegetation degradation amid anthropogenic activities, habitat loss, over-exploitation, invasive plant species, and climate change [4].

Honey bee colonies contribute greatly to human well-being by contributing to the provision of ecosystem services and completing plants’ life cycles [5,6,7]. Although it is known that the ecosystem contributes positively to the sustainable development of life [8,9], there are no studies confirming the shared contribution of bees to achieve sustainable development goals through the ecosystem. Numerous research endeavors have demonstrated the importance of insect pollinators in achieving multiple sustainable development goals by regulating natural cycles, biological control, pollination, seed dispersal, and even as a biological inspiration [5,10,11]. The role of bees in pollinating plants has been emphasized as a direct contribution to food security and biodiversity. Therefore, sustainable development of honey bee resources and forage starts by identifying and documenting the honey bee plants to assure honey bees’ services.

There has been a constant demand for honey bee services since ancient times; however, the honey production sub-sector, which relied on limited floral resources could not fulfill the long-standing demand for honey and related products and byproducts while the country continued importing from abroad. Saudi Arabia currently imports more than 15,000 metric tons of table honey each year. Australia, Turkey, Mexico, Argentina, Pakistan, United States, Germany, and Yemen are the largest honey-producing countries, in order of volume imported to the country. However, this trend shall be changed and the country shall focus on boosting its apicultural production and declare self-dependency by advocating beekeeping development and apicultural resource development endeavors [2]. In doing so, rehabilitation and restoration of apicultural landscapes through reforestation and plantation activities while emphasizing local honey bee floral genetic resources shall draw the attention of development agents. However, so far, information about the high-value nectar and pollen sources, which include indigenous and locally available plant genetic material, is not largely available; when it is available, it is only limited to a few species [3]. Logically, it is worth identifying and documenting the honey bee plant species and creating an area-specific checklist to make use of the resources in times of development needs as part of the beekeeping development effort. The current list of honey bee flora of Al-Baha might not be the most comprehensive of all bee plant species, but it can at least be a foundation list of available honey bee flora for the government-community partnership and non-governmental organizations (NGOs) interested in rehabilitating the honey bee floral resource of this region.

Coupled with the mapping of botanical communities of bee plants in the region, the list can be an excellent developmental aid that contributes to improving the beekeeping conditions and increasing the average output production per colony. Therefore, this study offers a way for future restoration and rehabilitation works to boost the apicultural sub-sector and hence contributes to achieving some of the nation’s development goals, such as Vision 2030. In line with this, the current study is aimed at documenting honey bee flora in arid and semi-arid areas by enlisting bee-plant species in the Al-Baha Region of Saudi Arabia, respective to the different seasons.

## 2. Results

Pollen sources were found to be greater in number than the nectar and propolis source plants with recorded numbers of species being 122, 92, and 10, respectively. Similarly, plant family numbers were higher for pollen sources than the nectar and propolis, with 31, 14, and 3 plant family classes, respectively. In the current study, about 62 families were recorded as a source of nectar, pollen, and both nectar and pollen source plants. Asteraceae has the highest number of individual plant species with 35 in total, and all of which are considered as sources of nectar, pollen, or both. The Fabaceae family followed the Asteraceae in bee plant documentation in the region, while the Lamiaceae, Malvaceae, and the Amaranthaceae were found to contribute significantly, in this order of importance, to honey bee resources in the form of nectar and pollen. Euphorbiaceae, Plantaginaceae, Boraginaceae, Brassicaceae, and Acanthaceae were also among the top ten plant families holding major shares in the contribution of honey bee resources in the region. These ratings were based on the number of individual plant species contributing to the bee forage resource in the region. (Table 1 and Table 2).

Season-wiseanalysis of the distribution of bee plant species found that spring is the most valuable (*p* < 0.05) flowering season for the bees, followed by winter, autumn, and summer with 170, 128, 96, and 89 flowering plants, respectively. Spring and winter are more precious flowering seasons (*p* < 0.05) than autumn and summer for beekeepers in Al-Baha region, and the number of bee plants secreting nectar was 54 and 50, respectively (Table 3).

The variations in the distribution of flowering and not-flowering status of plant species were found to be significant (*p* < 0.05) in spring and winter, while only slight variations were seen between summer and autumn (Figure 1).

The percentages of plants that did not flower in summer and autumn were observed to be 64.16% and 68.28%, respectively (Table 3). The diversity index showed that the flowering plant species in spring (H′ = 2.22) and winter (H′ = 2.16) were more diverse than the autumn (H′ = 1.99) and summer (H′ = 1.94).

On the other hand, spring and winter were found to be the most valuable seasons with a high percentage of plants that flower at 63.43% and 45.48%, respectively (Table 4).

Results from the contingency analysis of the Chi-square test revealed that flowering plants behaved significantly (*p* < 0.05) differently distributed across all the seasons. A higher number of flowering plants were found to be significant in spring and winter with a value of *p* < 0.05. There is a significant (*p* < 0.05) clustering in the number of flowering plants in spring and winter compared with summer and autumn, as shown in the correspondence analysis (Figure 1).

Results of the current study show that wild plant species constitute 84.70%, followed by regional endemic (10.45%), near-endemic (2.99%), and endemic plants (1.87%) of the bee flora of the region. Many bee-forage plants like *Ziziphus spina-christi*, *Vechilia* species *Senegalia asak*, and *Senegalia hamulosa* are designated as rare and endangered. Some plant species are rare, such as *Blepharis edulis* and *Hypoestes forskaolii,* and are considered valuable honey sources with high rates (Figure 2), (Table 2).

## 3. Discussion

The current study is aimed at documenting honey bee flora in arid and semi-arid areas by enlisting honey bee plant species in Al-Baha Region of Saudi Arabia, respective of the different seasons. The existing flora in the current study area benefits the honey bees by providing forage resources, namely nectar, pollen, and propolis; ultimately, they were found to be the main rewards offered by flowers to honey bees. Meanwhile, pollen sources outweighed the nectar and propolis sources in the number of plant species. In agreement with the current study, plants certainly provide pollen more than nectar [12]. Based on the honey bee’s requirement for forage, adult bees mostly consume more nectar than pollen, whereas larvae need more volume of both pollen and nectar (bee bread) [13,14,15,16]. This fact has been proven in the current study, as well as a sustainable mutualism association between the bee and plant species, where bees benefit the plants through pollination while they obtain their basic nutrients from the pollen. Honey bees do not gather the forage resource for their survival only, but for their broods also [17,18,19], which guarantees the continuation of their species.

Similar to the current study, Al-Ghamdi et al. [3] mentioned that bee plants are important in strengthening bee colonies, and mentioned the total number of bee plants found in the Al-Baha Region is 204 plant species under 58 families dominated by nine main honey source plants. The distribution and availability of pollen source plants could be attributed to the fact that the genetic constitution of plants often produces pollen grain for sexual multiplication, and bees benefit from the process of pollination whilst collecting pollen. Understanding the floral resource (nectar, pollen, and propolis) and their flowering season is critically important for the improvement of the management of carrying capacity in their habitats [20]. Similar studies have implied that the flowering calendar varies among species and locations. However, listing and classifying plants based on their potential benefit to honey bees is essential to help support beekeeping [21].

Similar results were reported by Al-Namazi et al. [22]. They found that the plant diversity in southwest Saudi Arabia was about 319 plant species from 75 families and 228 genera, among which only two species are endemic; 14 are endemic to the Arabian Peninsula, five are regional-endemic only found in East Africa and the Arabian Peninsula, and 39 are rare and endangered. The significant number of rare and endangered plant species could raise an alarm to initiate effective remedial measures for the preservation and conservation of the existing flora. Shada Mountain, the highest landscape in the lowlands of Al-Baha Region of southwestern Saudi Arabia, is the most important plant area and was found to have 495 plant species belonging to 314 genera and 76 families, including 19 endemic species and 43 endangered species, accounting for 22% of Saudi Arabia’s total flora [23].

Furthermore, the outcomes of this study showed that about 268 plant species under 62 families were recorded in the study area. They are there for providing bee floral rewards, pollen, nectar, and propolis either in a combined or separate manner in that order of importance of contribution to the wellbeing of honey bees. Similar floral richness to the current study related to different geographic, edaphic, and environmental factors was also reported by Al-Aklabi et al. [24]. The fact is that the research location has varied geographical and climate elements that could explain the variance in the diversity of flowering plants throughout time. The most plant families observed in the region were the Asteraceae, Fabaceae, Lamiaceae, and Malvaceae. These families were found to have the highest number of species in Saudi Arabia, likely due to their long-developed stress and drought tolerance abilities. Similarly, Amaranthaceae, Euphorbiaceae, Plantaginaceae, Boraginaceae, Brassicaceae, and Acanthaceae are known for their adaptation to arid and semi-arid regions, thus holding major shares in the honey bee resource contribution in the current region, which is mainly characterized by dry climates. Similar findings were reported by Kuppler et al. [25] and Zurbuchen and Müller [26].

In our study, spring and winter were relatively good seasons for honey bees in terms of pollen, nectar, and propolis availability. This might be due the relatively conducive plant growing weather conditions in winter and spring. The region’s erratic foggy weather in winter and early spring and relatively moderate temperature and rainfall records were reported by National Center for Metrology [27]. Autumn season is frequently coupled with heavy rain and fog, which is still considered favorable for most plants in highland locations and foothills. The impact of environmental conditions on species diversity and richness was reported by Slezák et al. [28] and Al Zandi et al. [29].

## 4. Materials and Methods

### 4.1. Study Sites

The study was conducted in the Al-Baha Region, southwestern Saudi Arabia starting from 2019 to the end of 2021. The ecological area consists of four regions, namely, the Tehama coastal area, the Tehama Foothills, the high-altitude area, and the eastern foothills [24]. The average temperature in the summer ranged between 20–30 °C and 8–20 °C in the winter. Rainfall varies among geographical areas. In the highlands, it ranged from 300 to 550 mm/year, while it was less than 150 mm/year in the Tehama Region. The humidity ranged from 40% to 58% from November to May [27]. The region is known for its erratic foggy weather in winter and early spring while the plant populations thrive well as relatively moderate temperature and rainfall records are common in these seasons. In May, the environmental and geographical characteristics create a favorable ecological niche for diversifying the life forms of plant species. Al-Baha Region has been described in many studies as one of the best ecological landscapes for harboring vast floral and faunal diversity in Saudi Arabia [3,22,27,28]. The sampled sites were mapped using ArcGIS software (Figure 3).

### 4.2. Sampling and Bee Plant Identification

The study concentrated on surveying the bee plant species distributed throughout the region for three consecutive years, from Jan 2019 to Dec 2021. A sampling model was used in many studies on enumerating plant diversity [24,30,31]. The survey covered an area of 180,000 m^2^ using a 20 × 20 m quadrat laid purposively to exhaustively include the most possible plant diversity. A total of 450 plots were considered. In areas with homogeneous plant cover, samples were taken randomly, leading to the use of purposive random sampling technique. Plant specimens were collected from the field and immediately pressed to get dried and finally glued on herbarium sheets; then, all specimens were archived and placed in the herbarium of Al-Baha University based on the scientific methods of identification following Collenette [32].

The plant species valuable to bees were identified based on flower avaliability, bees’ action during foraging hours on the flowers, presence of pollen in the corbiculae, and the old regional beekeepers’ experience (who observe how the bees take resources from the flowers: when the bees send their proboscis to a flower, it implies that the plant is a nectar source. It is a pollen source if the bees uptake the pollen powder in their body and collect it in the pollen basket. It is also clear when bees collect the propolis and can be observed by the researchers [33]). After data collection, we classified the bee plants into different groups according to the type of collected forages as follows: nectar (N) source, pollen (P) source, and both pollen and nectar source plants. The latter group was marked according to the main dominance source as follows: N + P (the main source of the forage was nectar and some pollen was observed) or P + N (the main source of the forage was pollen and some nectar was observed). Careful observations were made on the bees’ actions while collecting resources for bee forage source plant categorization. The propolis (Po) source plants were determined by the observations made on bees’ visits to vegetative parts of plants collecting the plant secretions from leaves, stems, fruits, and branches (Appendix A: images showing bees on the forage plants). Observations on honey bee visits were conducted during foraging hours respective of the different seasons: every two hours from 5:30 AM to 6:30 PM in spring, summer and autumn, and every one hour from 9:00 AM to 5:00 PM in winter. In most parts of the region, the cold stresses of the winter season are common during early morning and late afternoon. Data were filtered and managed using Microsoft Office Excel 2013. The responses of nominal variables like the seasonal distribution of flowering plants in spring, summer, autumn, and winter were tested using Chi-square (χ^2^) tests of contingency analysis, followed by correspondence analysis to show the relationship between the variables. Shannon’s diversity index (*H*′) of flowering plants in different seasons was applied to test the species richness among the flowering seasons using Multivariate Statistical Package (MVSP) version 3.22 [34].

Shannon’s diversity index [35]
$$H^{\prime }=\sum_{i=1}^{s}piLn(pi)$$
where *pi* is the proportion of individuals of species in its species.

All statistical analysis was performed in JMP statistical software version 5 to analyze the variation among variables [36].

## 5. Conclusions

Al-Baha Region is found to be a potential place for beekeeping practices and producing honey in local as well as international markets. The high demand by many local consumers is an indicator of its high preference. The plant diversity and environmental factors could be the secret behind the quantity and quality of the Al-Baha honey. The conservation of honey bee plant species is critical due to the increase in genetic loss of important honey bee plant species from naturally existing forests. The narrow population of these economic plant species cast a shadow on the purchasing process of honey products and increased price rate. All the bee forage plants under danger and deterioration can be propagated and conserved using in situ and ex situ techniques. Apart from this, variation of plants in blooming seasons may support the continuous flow of honey throughout the year. For sustainable and continuous honey production, urgent action has to be taken to protect the bee forage plant populations in arid and semi-arid areas of Al-Baha, as the floral diversity is dwindling due to harsh ecosystem components. Furthermore, the current research outcome exposes the increasing demand to conserve the floral diversity through the anticipation of think tanks from all related sectors, including scientists in collaboration with government and non-government agencies. Moreover, awareness creates endeavors in conservation for the local inhabitants of the Kingdom of Saudi Arabia.

## Figures and Tables

**Figure 1 plants-12-01402-f001:**
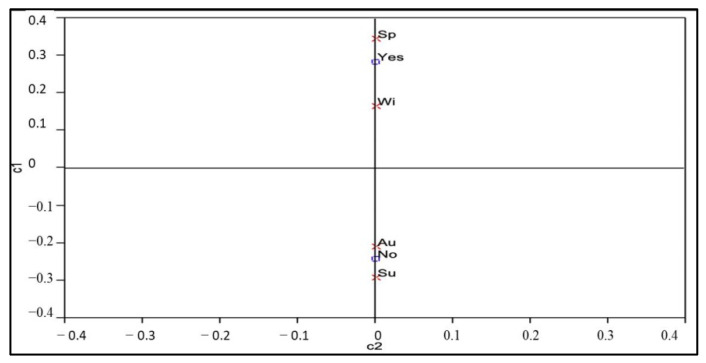
Correspondence analysis of the distribution of flowering honey bee plants across the different seasons. × = Seasons; 
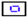
 = Flowering status; Sp = Spring; Wi = Winter; Au = Autumn; Su = Summer. The category “Yes” is quite near to “Winter” and “Spring” while “No” is related to “Autumn” and “Summer”, implying that flowering honey bee plants occur more during winter and spring and less in summer and autumn.

**Figure 2 plants-12-01402-f002:**
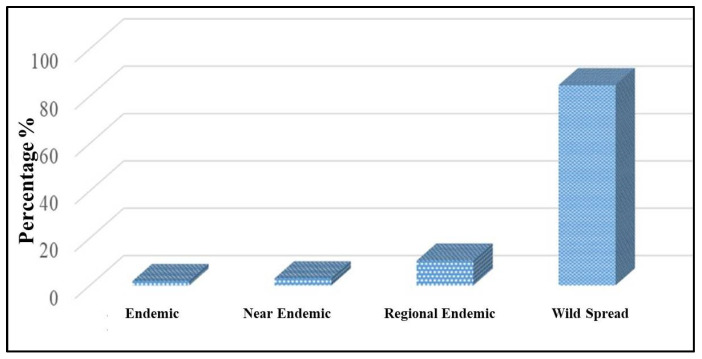
Percentage of endemism among bee plant species in Al-Baha region, southwestern Saudi Arabia.

**Figure 3 plants-12-01402-f003:**
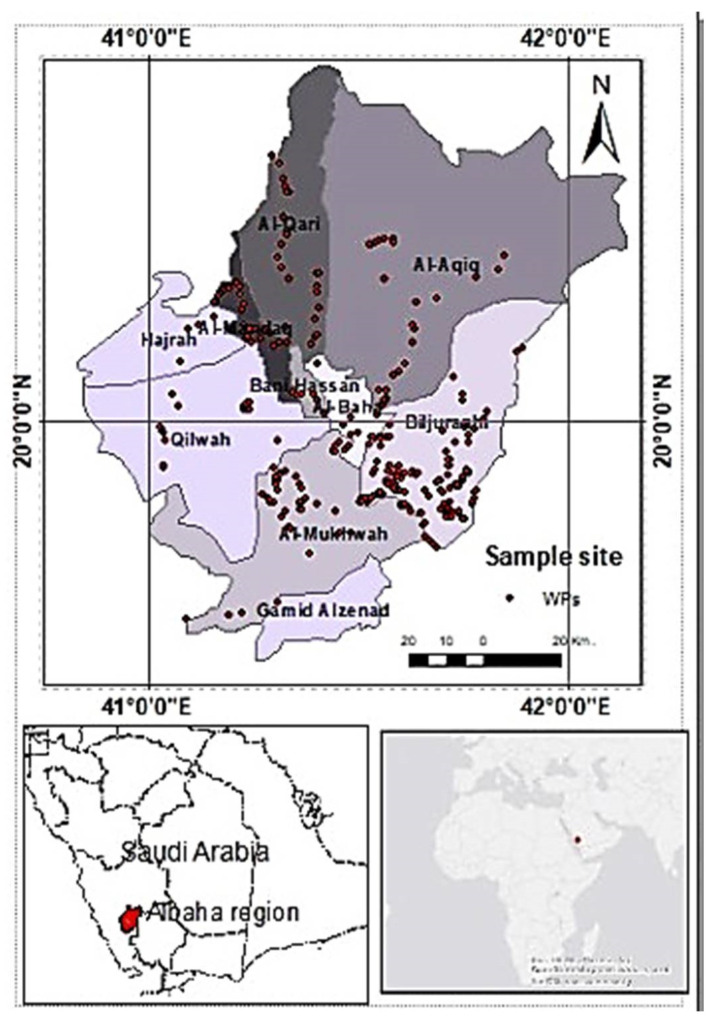
Map of the study area and locations of sample sites.

**Table 1 plants-12-01402-t001:** Number of plant species and families benefiting the bees in Al-Baha region, southwestern Saudi Arabia.

	N		N + P		P + N		P		Pro		N + P + Pro	
	No.	%	No.	%	No.	%	No.	%	No.	%	No.	%
Number of plants	92	25.1	109	29.8	28	7.7	122	31	10	3	5	2
Number of Families	14	15.4	29	31.9	12	13.2	33.3	34.1	2.7	3.3	1.4	2.2

N (Nectar), P (Pollen), Pro (Propolis).

**Table 2 plants-12-01402-t002:** Checklist of the bee forage plants from Al Baha, Southwestern Saudi Arabia. Nectar (N), Pollen (P), and Propolis (Pro).

Botanical Name	Family	Source of Forage	Flowering Calendar	Endemism
*Anisotes trisulcus* (Forssk.) Nees Barleria bispinosa	Acanthaceae Juss	N	Jan, Feb, Mar, Apr and May	Regional endemic
2. *Barleria bispinosa* (Forssk.) Vahl		N + P	Jan, Feb, Mar and Dec	Near-endemic
3. *Barlria prionitis* L.		N + P	Jan, Feb, Mar, Apr and May	Wild spread
4. *Blepharis edulis* (Forssk.) Pers		N	Jan, Feb, Mar, Apr and May	Wild spread
5. *Hypoestes forskaolii* (Vahl) R. Br. Justicia flava		N	Jan, Feb, Mar and Apr	Wild spread
6. *Justicia flava* (Forssk.) Vahl		N + P	Jan, Feb and Mar	Wild spread
7. *Justicia odora* (Forssk.) Lam. Leucas alba		N + P	flowering calendar; Jan, Feb, Mar and Dec	Wild spread
8. *Sesuvium portulacastrum* L.	Aizoaceae Martinov	N + P	Aug and Sept	Wild spread
9. *Trianthema portulacastrum* L.		N + P	Aug and Sept	Wild spread
10. *Achyranthes aspera* L.	Amaranthaceae Juss.	P	Mar, Apr and May	Wild spread
11. *Aerva javanica* (Burm.f.) Juss. ex Schult.		P	Jan, Feb, Mar, Apr, May, Nov and Dec	Wild spread
12. *Amaranthus blitum* L. (=*Amaranthus viridis* All.)		P	Sept, Oct and Nov	Wild spread
13. *Chenopodiastrum murale* (L.) S. Fuentes, Uotila & Borsch (=*Chenopodium murale* L.)		N	Dec and Jan	Wild spread
14. *Chenopodium album* L.		N	Dec and Jan	Wild spread
15. *Digera muricata* (L.) Mart.		N	Jan, Feb and Mar.	Wild spread
16. *Dysphania ambrosioides* (L.) Mosyakin & Clemants (=*Chenopodium ambrosioides* L.)		N	Oct, Nov and Dec	Wild spread
17. *Oxybasis glauca* (L.) S. Fuentes, Uotila & Borsch (=*Chenopodium glaucum* L.)		N	May, June and July	Wild spread
18. *Psilotrichum gnaphalobryum* (Hochst.) Schinz		N	Dec and Jan	Wild spread
19. *Salsola kali* L.		N	Nov and Dec	Wild spread
20. *Salsola tragus* L.		P	Mar, Apr, May, and June	Wild spread
21. *Crinum album* (Forssk.) Herb	Amaranthaceae Juss.	P	Feb and Mar	Endemic
22. *Searsia retinorrhoea* (Steud. ex Oliv.) Moffett	Anacardiaceae R.Br.	N	Aug, Sept, Oct and Nov	
23. *Ammi majus* L.	Apiaceae Lindl.	N + P	Apr, May and June	
24. *Conium maculatum* L.		N + P	Aug and Sept	
25. *Foeniculum vulgare* Mill		N + P	Apr, Dec, Jan, Feb and Mar.	
26. *Calotropis procera* (Aiton) W. T. Aiton	Apocynaceae Juss.	N + P + Pr	Feb, Mar and Nov	Wild spread
27. *Carissa spinarum* L.		N	Apr. May, June, July, Aug. and Sept.	Wild spread
28. *Leptadenia pyrotechnica* (Forssk.)		N	Nov. and Dec.	Wild spread
29. *Cynanchum viminale* (L.) L. (=*Sarcostemma viminale* (L.) R. Br.		N	Apr. May, June, July, Aug. and Sept.	Wild spread
30. *Phoenix caespitosa* Chiov.	Arecaceae Bercht. & J.Presl	N	Nov. and Dec.	Wild spread
31. *Phoenix dactylifera* L.		N	Feb. Mar. and Apr.	Wild spread
32. *Asparagus africanus* Lam.	Asparagaceae Juss	N + P	Feb. Apr. May, Nov. and Dec	Wild spread
33. *Aloe castellorum* J. R. I. Wood	Asphodelaceae Juss	N + P	Feb, Mar, Apr, May, June, July and Aug	Near-endemic
34. *Aloe pseudorubroviolacea* Lavranos & Collen.		N + P	Apr and May	Endemic
35. *Aloe shadensis Lavranos* & Collen.		N + P	Apr and May	Endemic
36. *Aloe saudiarabica* T. A. McCoy		N + P	Mar, Apr and May	Endemic
37. *Asphodelus fistulosus* L.		P	May and June	Wild spread
38. *Asphodelus tenuifolius* Cav.		P	Jan, Feb, Mar, Apr and May	Wild spread
39. *Achillea arabica Kotschy* (=Achillea biebersteinii Afanasiev)	Asteraceae Bercht. & J.Presl	P	Mar. and Apr.	Wild spread
40. *Ambrosia maritima* L.		N	Aug, Sept, Oct and Nov	Wild spread
41. *Baccharoides schimperi* (DC.) Isawumi, El-Ghazaly & B. Nord. (=*Vernonia schimperi* DC.)		N	May and June	Wild spread
42. *Bidens pilosa* L.		N + P	Sept, Oct, Nov and Dec	Wild spread
43. *Calendula arvensis* L.		N + P	Jan, Feb, Mar and Apr.	Wild spread
44. *Carduus pycnocephalus* L.		N + P	Mar, Apr and May	Wild spread
45. *Centaurea sinaica* DC.		N + P	Jan, Feb, Mar, Apr, May and Dec	Wild spread
46. *Centaurea schimperi* DC.		N + P	Jan, Feb, Mar, Apr, Nov and Dec	Wild spread
47. *Cichorium bottae* Deflers		N + P	Dec and Jan	Near-endemic
48. *Cichorium intybus* L.		N + P	May, June, July and Aug	
49. *Erigeron incanus* Vahl (=*Conyza incana* (Vahl) Willd.)		N + P	Nov, Dec and Jan	Wild spread
50. *Eschenbachia gouanii* (L.) G. L. Nesom (=*Conyza hochstetterii* Sch.Bip. ex A.Rich.)		N + P	Jan and Feb	Wild spread
51. *Eschenbachia stricta* (Willd.) Raizada. (=synonym *Conyza stricta* Wall.)		N + P	Nov, Dec and Jan	Wild spread
52. *Crepis foetida* L.		P	Apr and May	Wild spread
53. *Crepis rueppellii* Sch.Bip.		P	Jan and Feb.	Regional endemic
54. *Echinops polyceras* Boiss		N	Apr, May, June and July	Wild spread
55. *Felicia abyssinica* Sch.Bip. ex A. Rich.		N + P	Dec, Jan, Feb, Mar, Apr and May	Regional endemic
56. *Felicia dentata* (A.Rich.) Dandy		N + P	Dec, Jan, Feb, Mar, Apr, May and June	Regional endemic
57. *Flaveria trinervia* (Spreng.) C. Mohr		P + N	Apr, May, June, July, Dec and Jan.	Wild spread
58. *Helichrysum glumaceum* DC.		N + P	Nov, Dec, Jan, Feb, Mar, Apr and May	Wild spread
59. *Lactuca serriola* L		P + N	July, Aug and Sept.	Wild spread
60. *Onopordum heteracanthum* C. A. Mey		N	Apr, May and June	Wild spread
61. *Picris asplenioides* subsp. asplenioides (=Picris radicata Less.)		N + P	May and June	Wild spread
62. *Pluchea dioscorides* (L.) DC.		N + Pro	Nov, Dec, Jan, Feb, Mar and Apr	Wild spread
63. *Psiadia punctulata* Vatke		N + P	Nov, Dec, Jan, Feb and Mar.	Wild spread
64. *Pulicaria undulata* (L.) C.A.Mey.		N + P	Dec, Jan, Feb, Mar, Apr, May and June	Wild spread
65. *Pulicaria petiolaris* Jaub. & Spach		P + N	Dec, Jan and Feb	Wild spread
66. *Pulicaria schimperi* DC		P + N	Dec, Jan and Feb	Wild spread
67. *Ramaliella musilii* (Velen.) Zaika, Sukhor. & N.Kilian (=*Scorzonera musilii Velen*)		N + P	Mar, Apr, May and June	Wild spread
68. *Sonchus oleraceus* L.		P	Jan, Feb, Mar, Apr, May and June	Wild spread
69. *Osteospermum vaillantii* (Decne.) Norl		N + P	Jan, Feb, Mar, Apr and May	Wild spread
70. *Tagetes minuta* L.		N + P	Apr and May	Wild spread
71. *Verbesina encelioides* (Cav.) Benth. & Hook.f. ex A.Gray		N + P	Jan, Feb, Mar, Apr and May	Wild spread
72. *Orbivestus cinerascens* (Sch.Bip.) H.Rob. (=*Vernonia cinerascens* Sch.Bip.)		N + P	Aug and Sept.	Wild spread
73. *Veronica anagallis-aquatica* L.		N + P	Dec, Jan and Feb	Wild spread
74. *Cordia africana* Lam.	Boraginaceae Juss.	N	Jun, July and Aug	Wild spread
75. *Cordia monoica* Roxb.		N	Jun, July and Aug	Wild spread
76. *Echium rauwolfii* Delile		N + P	Jan, Feb, Mar, Apr and May	Wild spread
77. *Echium* sp		N + P	Jan, Feb and Mar	Wild spread
78. *Ehretia obtusifolia Hochst*. ex A.DC		N	Aug, Sept and Oct	Wild spread
79. *Heliotropium arbainense* Fresen		N + P	Jan, Feb, Mar, Apr, Aug, Sept and Oct	Wild spread
80. *Heliotropium longiflorum* (A.DC.) Jaub. & Spach		N + P	July, Aug, Sept and Oct	Wild spread
81. *Heliotropium pterocarpum* (DC.) Hochst. & Steud. ex Bunge		N	Aug, Sept, Oct, Nov, Dec, Jan, Feb and Mar	Wild spread
82. *Coincya tournefortii* (Gouan) Alcaraz, T.E.Díaz, Rivas Mart. & Sánchez-Gómez (=*Brassica tournefortii* Gouan)	Brassicaceae Burnett	P + N	Feb, Mar and Apr	Wild spread
83. *Crambe orientalis* L.		N + P	Nov, Dec, Jan and Feb	Wild spread
84. *Eruca vesicaria* (L.) Cav. (=Eruca sativa Mill.)		N + P	Nov, Dec, Jan, Feb, Mar and Apr.	Wild spread
85. *Nasturtium officinale* W.T.Aiton		P	Nov, Dec, Jan, Feb, and Mar	Wild spread
86. *Rapistrum rugosum* (L.) All.		P	Aug, Sept, Oct. Nov and Dec	Wild spread
87. *Sinapis alba* L.		N + P	Dec, Jan, Feb and Mar	Wild spread
88. *Sisymbrium irio* L.		P + N	Jan, Feb and Mar	Wild spread
89. *Sisymbrium orientale* L		P + N	Jan, Feb and Mar	Wild spread
90. *Commiphora gileadensis* (L.) C.Chr.	Burseraceae Kunth	P	Jan, Feb, Mar and Apr	Regional endemic
91. *Commiphora kataf* (Forssk.) Engl.		P	Nov, Dec, Jan and Feb	Regional endemic
92. *Commiphora kua* (R.Br. ex Royle) Vollesen		P	Nov, Dec, Jan and Feb	Regional endemic
93. *Commiphora myrrha* (T.Nees) Engl.		P	Oct, Nov and Dec	Regional endemic
94. *Opuntia ficus-indica* (L.) Mill	Cactaceae Juss	N + P	May and June	Wild spread
95. *Cylindropuntia imbricata* subsp. *rosea* (DC.) M.A.Baker		N + P	May and June	Wild spread
96. *Celtis africana* Burm.f.	Cannabaceae Martinov	P	Feb and Mar	Wild spread
97. *Boscia integrifolia* J.St.-Hil.	Capparaceae Juss	N	Aug and Sept	Wild spread
98. *Capparis cartilaginea* Decne		P	Apr, May, June, July and Aug	Wild spread
99. *Capparis decidua* (Forssk.) Edgew		P	Oct and Nov	Wild spread
100. *Capparis tomentosa* Lam		P	Mar and Apr	Wild spread
101. *Maerua crassifolia* Forssk		P + N	July and Aug	Wild spread
102. *Maerua oblongifolia* (Forssk.) A.Rich		P + N	May, June and July	Wild spread
103. *Polycarpaea repens* (Forssk.) Asch. & Schweinf	Caryophyllaceae Juss	P	Jan, Feb and Mar	Wild spread
104. *Gymnosporia parviflora* (Vahl) Chiov. (=*Maytenus parviflora* (Vahl) Sebsebe)	Celastraceae R.Br.	N + P	Sept, Oct and Nov	Wild spread
105. *Cleome pallida* Kotschy (=*Dipterygium glaucum* Decne.)	Cleomaceae Airy Shaw	P + N	Mar and Apr	Wild spread
106. *Cleome ramosissima* Parl. ex Webb		P + N	Nov and Dec	Regional endemic
107. *Cleome gynandra* L.		N	Aug, Sept and Oct	Wild spread
108. *Combretum pisoniiflorum* (Klotzsch) Engl. (=*Combretum molle* R.Br. ex G.Don)	Combretaceae R.Br	P + N	Mar, Apr and May	Wild spread
109. *Combretum aculeatum* Vent		N + P	Feb and Mar and Apr	Wild spread
110. *Commelina albescens* Hassk	Commelinaceae Mirb	P	Jan, Jan, Fab and Mar	Wild spread
111. *Commelina Africana* L.		P	Jan, Jan, Fab and Mar	Wild spread
112. *Commelina africana* subsp. Africana		P	Jan, Jan, Fab and Mar	Wild spread
113. *Convolvulus arvensis* L.	Convolvulaceae Juss	N	June, July, Aug and Sept.	Wild spread
114. *Convolvulus asyrensis* Kotschy		N	Jan, Feb and Mar	Endemic
115. *Ipomoea obscura* (L.) Ker Gawl.		N	Oct and Nov	Wild spread
116. *Crassula schimperi* Fisch. & C.A.Mey	Crassulaceae J.St.-Hil.	P	Jan and Feb	Wild spread
117. *Citrullus colocynthis* (L.) Schrad	Cucurbitaceae Juss	P	Jan, Feb, Jun and Oct	Wild spread
118. *Juniperus procera* Hochst. ex Endl	Cupressaceae Gray	Pro	Resin secretion Jan, Feb, Mar and Apr.	Regional Endemic
119. *Diospyros mespiliformis* Hochst. ex A.DC.	Ebenaceae Gürke	N	Apr, Aug and Sept	Wild spread
120. *Euclea racemosa* L		N + P	Oct and Nov	Wild spread
121. *Erica arborea* L.	Ericaceae Durande	N + P	Mar, Apr and May	Wild spread
122. *Euphorbia balbisii* Boiss. (=*Euphorbia serpens* Balb. ex Boiss.)	Euphorbiaceae Juss	N	Dec, Jan, Feb and Mar	Wild spread
123. *Euphorbia cuneata* Vahl		N	July and Aug	Wild spread
124. *Euphorbia falcata* L.		N	Dec and Jan	Wild spread
125. *Euphorbia inarticulata* Schweinf		N	Dec, Nov, Jan, Feb, Mar and Apr	Near-endemic
126. *Euphorbia parciramulosa* Schweinf.		N	Mar and Apr	endemic
127. *Euphorbia schimperiana* var. schimperiana Scheele		N	July and Aug	Wild spread
128. *Jatropha glauca* Vahl		N	Aug, Sept and Oct.	Region Endemic
129. *Jatropha pelargoniifolia* Courbon		N	Mar and Apr	Region Endemic
130. *Ricinus communis* L.		N	Jan, Feb, Mar, Apr, May and June	Region Endemic
131. *Argyrolobium arabicum* (Decne.) Jaub. & Spach	Fabaceae Lindl	N	Dec, Jan, Feb and Mar.	Wild spread
132. *Astragalus atropilosulus* subsp. *atropilosulus* (=*Astragalus atropilosulus* subsp. *abyssinicus* (Hochst.)) Gillett		P	Dec, Jan, Feb, Mar, Apr and May	Region Endemic
133. *Astragalus vogelii* subsp. *fatmensis* (Hochst. ex Chiov.) Maire (=*Astragalus fatmensis* Hochst. ex Chiov)		P	Dec, Jan, Feb and Mar	Wild spread
134. *Crotalaria emarginella* Vatke		P	Jan Feb, Mar, Apr, Sept and Oct.	Region Endemic
135. *Delonix elata* (L.) Gamble		P	Jan, Feb, Mar and Apr	Region Endemic
136. *Dorycnopsis abyssinica* (A.Rich.) V.N.Tikhom. & D.D.Sokoloff (=*Vermifrux abyssinica* (A.Rich.) J.B.Gillett)		P + N	Jan, Feb and Mar	Wild spread
137. *Faidherbia albida* (Delile) A.Chev		P	Apr, May, Sept and Oct	Wild spread
138. *Lotus quinatus* (Forssk.) Gillent		N	Jan, Feb, Mar, Apr and May	Wild spread
139. *Medicago laciniata* (L.) Mill		N	Nov, Dec, Jan, Feb and Mar.	Wild spread
140. *Medicago minima* (L.) Bartal		N	Mar and Apr	Wild spread
141. *Medicago polymorpha* L		N	Dec, Jan, Feb and Mar	Wild spread
142. *Melilotus indicus* (L.) All.		N	Dec, Jan, Feb and Mar	Wild spread
143. *Onobrychis ptolemaica* (Delile) DC.		N	Jan, Feb, Mar and Apr	Wild spread
144. *Rhynchosia malacophylla* (Spreng.)		N	July and Aug	Wild spread
145. *Senegalia asak* (Forssk.) Kyal. & Boatwr. (=*Acacia asak* (Forssk.) Willd		N + P	Apr, May, June, July, Aug and Sept.	Wild spread
146. *Senegalia hamulosa* (Benth.) Boatwr. (=*Acacia hamulosa* Benth.)		N + P	Apr, May, June, July and Aug	Wild spread
147. *Senna alexandrina* Mill		P	Apr, May and Nov	Wild spread
148. *Senna italica* Mill		P	Apr, May and Nov	Wild spread
149. *Tephrosia nubica* (Boiss.) Baker		N	Nov and Dec.	Wild spread
150. *Trifolium arvense* L.		N + P	Mar, Apr and May	Wild spread
151. *Trifolium campestre* Schreb		N + P	Mar and Apr.	Wild spread
152. *Trifolium retusum* L.		N	Mar, Apr and May	Wild spread
153. *Vachellia etbaica* (Schweinf.) Kyal. & Boatwr. (=Acacia etbaica Schweinf)		N + P	Apr, May, June, July, Aug and sept.	Wild spread
154. *Vachellia flava* (Forssk.) Kyal. & Boatwr. (=*Acacia ehrenbergiana* Heyne)		N + P	Feb, Mar, Apr, May, June, July and Aug.	Wild spread
155. *Vachellia gerrardii* (Benth.) P.J.H.Hurter (=*Acacia gerrardii* Benth)		N	May, June, July, Aug, sept and Oct.	Wild spread
156. *Vachellia johnwoodii* (Boulos) Ragup.، Seigler، Ebinger & Maslin (=*Acacia johnwoodii* Boulos)		N + P	July, Aug, sept, Oct, Nov, Dec, Jan and Feb	Wild spread
157. *Vachellia oerfota* (Forssk.) Kyal. & Boatwr = (*Acacia oerfota*)		N	sept and Oct.	Wild spread
158. *Vachellia origena* (Hunde) Kyal. & Boatwr. (=*Acacia origena* Hunde)		N	Apr, May and June	Wild spread
159. *Vachellia tortilis* (Forssk.) Galasso & Banfi (=*Acacia tortilis* (Forssk.) Heyne)		N	Jan, Feb, Mar, Apr, Oct, Nov and Dec.	Wild spread
160. *Erodium cicutarium* (L.) L’Hér	Geraniaceae Juss	N	Dec, Jan and Feb	Wild spread
161. *Erodium malacoides* (L.) L’Hér		N	Dec, Jan and Feb	Wild spread
162. *Erodium neuradifolium* Delile ex Godr		N	Dec, Jan and Feb	Wild spread
163. *Geranium molle* L.		N + P	Apr and May	Wild spread
164. *Pelargonium multibracteatum* Hochst. ex A.Rich.		N + P	May and June	Regional Endemic
165. *Gladiolus dalenii* Van Geel	Iridaceae Juss	N + P	Jan, Feb and Mar	Wild spread
166. *Isodon ternifolius* (D.Don) Kudô (=Plectranthus ternifolius D.Don)	Lamiaceae Martinov	N	Feb, Mar, Apr and May and July	Wild spread
167. *Lavandula atriplicifolia* Benth		N	May, June, Sept and Oct	Wild spread
168. *Lavandula citriodora* A.G.Mill.		N	May, June, Sept, Oct and Nov	Near-endemic
169. *Lavandula coronopifolia* Poir		N	Dec and Jan	Wild spread
170. *Lavandula dentata* L.		N	Nov, Dec, Jan, Feb, Mar, Apr and May	Wild spread
171. *Leucas alba* (Forssk.) Sebald		N + P	Apr, May, July, Aug and Sept.	Near-endemic
172. *Leucas glabrata* (Vahl) Sm.		N	Sept, Oct, Nov, Dec, Jan, Feb, Mar, Apr and May	Wild spread
173. *Mentha longifolia* L.		N	Oct, Nov, Dec, Mar, Apr and May	Wild spread
174. *Micromeria imbricata* (Forssk.) C.Chr.		N	Dec, Nov, Jan, Feb and Mar.	Wild spread
175. *Nepeta deflersiana* Schweinf. ex Hedge		N	Jan, Feb, Mar, Apr, May and June	Near-endemic
176. *Ocimum filamentosum* Forssk.		N + P	Mar, Apr, May and June	Wild spread
177. *Ocimum forskoelei* Benth.		N + P	Jan and July	Wild spread
178. *Otostegia fruticosa* (Forssk.) Schweinf. ex Penzig		N	Jan, Feb, June, Sept, Oct and Nov	Wild spread
179. *Premna resinosa* (Hochst.) Schauer		N	Mar, Apr, May and June	Wild spread
180. *Coleus arabicus* Benth. (=*Plectranthus asirensis* J.R.I.Wood)		N	Jan, Feb and Mar.	Wild spread
181. *Salvia aegyptiaca* L.		N	Apr, May and June.	Wild spread
182. *Salvia dianthera* Roth (=*Meriandra bengalensis* (J.Koenig ex Roxb.) Benth.)		N	Jan, Feb, Mar, Apr and May	Wild spread
183. *Salvia merjamie* Forssk.		N	Mar and Apr	Regional Endemic
184. *Teucrium yemense* Deflers		N + P	Feb, Mar, Apr and May	Regional Endemic
185. *Lawsonia inermis* L.	Lythraceae J.St.-Hil.	P	Nov, Dec and Jan	Wild spread
186. *Corchorus olitorius* L.	Malvaceae Juss	P	Nov, Dec, Jan and Feb	Wild spread
187. *Grewia erythraea* Schweinf		P + N	May and Oct.	Wild spread
188. *Grewia tembensis* Fresen		P + N	Apr, May, Sept and Oct.	Wild spread
189. *Grewia tenax* (Forssk.) Fiori		P + N	Aug, Sept, Oct and Nov	Wild spread
190. *Grewia trichocarpa* Hochst. ex A.Rich.		P + N	May, Sept and Oct	Regional Endemic
191. *Grewia mollis* Juss. (=*Grewia velutina* Franch.)		P + N	Aug and Sept.	Wild spread
192. *Grewia villosa* Willd		P + N	Sept and Oct	Wild spread
193. *Hibiscus aponeurus* Sprague & Hutch		N + P	Aug, Sept, Oct, Nov and Dec	Regional Endemic
194. *Hibiscus deflersii* Schweinf. ex Cufod		N + P	Dec, Apr and May	Regional Endemic
195. *Hibiscus micranthus* L.f.		N + P	Apr and Sept.	Wild spread
196. *Hibiscus vitifolius* L.		P	Jan and Feb	Wild spread
197. *Malva parviflora* L.		P	Nov and Sept.	Wild spread
198. *Triumfetta heterocarpa* Sprague & Hutch.		P	Nov, Dec, Jan and Feb	Wild spread
199. *Glinus lotoides* L.	Molluginaceae Bartl	N	Dec and Jan	Wild spread
200. *Moringa peregrina* (Forssk.) Fiori	Moringaceae Martinov	P	Apr and May	Wild spread
201. *Ficus carica* L.	Moraceae Gaudich	Pro	Resin secretion Mar, Apr, May and June	Wild spread
202. *Ficus glumosa* Delile		Pro	Resin secretion Sept, Oct, Nov and Dec	Wild spread
203. *Ficus ingens* (Miq.) Miq		Pro	Resin secretion Oct, Nov, Dec and Jan	Wild spread
204. *Ficus palmata Forssk*		Pro	Resin secretion Mar, Apr, May and June	Wild spread
205. *Ficus salicifolia* Vahl (=*Ficus cordata* subsp. *salicifolia* (Vahl) C.C.Berg)		Pro	Resin secretion Nov, Dec and Jan	Wild spread
206. *Ficus sycomorus* L.		Pro	Resin secretion Nov, Dec, Jan and Feb	Wild spread
207. *Ficus vasta* Forssk		Pro	Mar, Apr and May	Regional Endemic
208. *Boerhavia elegans* Choisy	Nyctaginaceae Juss	N	May and June	Wild spread
209. *Boerhavia diffusa* L.		N + P	Feb, Mar and Apr	Wild spread
210. *Ochna inermis* (Forssk.) Schweinf	Ochnaceae DC.	N + P	Mar, Apr, May, June, July and Aug.	Wild spread
211. *Jasminum grandiflorum* L.	Oleaceae Hoffmanns. & Link	N	Dec and Jan	Wild spread
212. *Olea europaea* L.		P + N	May and June	Wild spread
213. *Buchnera hispida* Buch.-Ham. ex D.Don	Orobanchaceae Vent	N	Sept and Oct	Wild spread
214. *Oxalis corniculata* L	Oxalidaceae R.Br.	N + P	July, Aug, Sept, Nov and Dec	Wild spread
215. *Argemone mexicana* L.	Papaveraceae Juss	P + N	Feb, Mar and Apr.	Wild spread
216. *Argemone ochroleuca* Sweet		P + N	Jan, Feb and Mar.	Wild spread
217. Glaucium corniculatum (L.) Curtis		P	Feb and Mar.	Wild spread
218. *Papaver decaisnei* Hochst. & Steud. ex Elkan		P	Feb and Mar	Wild spread
219. *Papaver dubium* L.		P	Feb and Mar	Wild spread
220. *Papaver laevigatum* M.Bieb. (=*Papaver dubium* subsp. laevigatum (M.Bieb.) Kadereit)		P	Feb and Mar	Wild spread
221. *Flueggea virosa* (Roxb. ex Willd.) Royle	Phyllanthaceae Martinov	P	Sept and Oct	Wild spread
222. *Anarrhinum forskaohlii* (J.F.Gmel.) Cufod	Plantaginaceae Juss	N	Jan, Feb, Mar and Apr	Wild spread
223. *Bacopa monnieri* (L.) Wettst		P	Nov and Dec	Wild spread
224. *Plantago afra* L.		P	Feb and Mar	Wild spread
225. *Plantago albicans* L. (=*Plantago ciliata* Boiss.)		P	Mar and Apr	Wild spread
226. *Plantago cylindrica* Forssk		P	Feb and Mar	Wild spread
227. *Plantago lanceolata* L.		P	Mar and Apr	Wild spread
228. *Schweinfurthia pterosperma* (A.Rich.) A.Braun		P	Feb and Mar	Wild spread
229. *Scoparia dulcis* L.		P	Jan and Feb	Wild spread
230. *Cenchrus ciliaris* L.	Poaceae Barnhart	P	Nov, Dec and Jan	Wild spread
231. *Cynodon dactylon* (L.) Pers.		P	Apr, May and June	Wild spread
232. *Polygala abyssinica* R.Br. ex Fresen	Polygalaceae Hoffmanns. & Link	P	May and June	Wild spread
233. *Polygala erioptera* DC.		P	Mar and Apr	Wild spread
234. *Polygala sinaica* var. glabrescens (Zohary) Boulos (=*Polygala negevensis* Danin)		P	Nov, Dec and Jan	Wild spread
235. *Polygala senensis* Klotzsch		P	Aug and Sept	Regional Endemic
236. *Rumex nervosus* Vahl	Polygonaceae Juss.	N + P	Jan, Feb, Mar and Apr	Wild spread
237. *Portulaca grandiflora* Hook	Portulacaceae Juss	P	Nov and Dec	Wild spread
238. *Portulaca kermesina* N.E.Br.		P	Nov, Dec and Jan	Wild spread
239. *Portulaca oleracea* L.		P + N	Feb, May and June	Wild spread
240. *Clematis hirsuta* Guill. & Perr	Ranunculaceae Juss	N + P	Feb, May and Apr	Wild spread
241. *Caylusea hexagyna* (Forssk.) M.L.Green	Resedaceae Martinov	P + N	Dec, Nov, Jan, Feb, Mar, Apr and May	Wild spread
242. *Ochradenus baccatus* Delile		P + N	Apr, May and June	Wild spread
243. *Rhamnus staddo* A.Rich	Rhamnaceae Juss	P + N	Apr and May	Regional Endemic
244. *Sageretia thea* (Osbeck) M.C.Johnst		N	Jan, July and Aug	Wild spread
245. *Ziziphus mucronata* Willd		N	Sept and Oct	Wild spread
246. *Ziziphus spina-christi* (L.) Desf.		N	July, Aug, Sept and Oct	Wild spread
247. *Rosa abyssinica* R.Br. ex Lindl	Rosaceae Juss	N + P	Apr and May	Wild spread
248. *Rubus creticus* Tourn. ex L.		N + P	May, June and July	Wild spread
249. *Pavetta gardeniifolia* var. gardeniifolia Hochst. ex A.Rich.(=*Pavetta longiflora*.)	Rubiaceae Juss	N	Mar, Apr, May and June	Wild spread
250. *Psydrax schimperianus* (A.Rich.) Bridson		N	Apr and May	Wild spread
251. *Pyrostria phyllanthoidea* (Baill.) Bridson		N	Apr and May	Regional Endemic
252. *Ruta chalepensis* L.	Rutaceae Juss	N	Jan, Feb, Mar, Apr and May	Wild spread
253. *Dodonaea viscosa* subsp. angustifolia (L.f.) J.G.West	Sapindaceae Juss	P	Jan, Feb and Mar	Wild spread
254. *Mimusops laurifolia* (Forssk.) Friis		P	June and July	Regional Endemic
255. *Buddleja polystachya* Fresen	Scrophulariaceae	P	Mar, Apr and May	Wild spread
256. *Lycium shawii* Roem. & Schult	Solanaceae Juss	P	Oct and Nov	Wild spread
257. *Solanum glabratum* Dunal		N	Nov, Dec and Jan	Wild spread
258. *Solanum incanum* L.		P + N	Apr, May, Sept, Oct, Nov, Dec and Jan	Wild spread
259. *Solanum schimperianum* Hochst (=*Solanum schimperianum* subvar. *cordifolium* Bitter)		P + N	Sept, Oct, Nov and Dec	Regional Endemic
260. *Solanum villosum* Mill		P+N	Oct, Nov, Dec, Feb, May and June	Wild spread
261. *Withania somnifera* (L.) Dunal		N + P	Dec, Jan, Feb, May and June	Wild spread
262. *Nuxia oppositifolia* (Hochst.) Benth	Stilbaceae Kunth	N	May, June and Sept.	Wild spread
263. *Tamarix aphylla* (L.) H.Karst	Tamaricaceae Link	P	May and June	Wild spread
264. *Tamarix nilotica* (Ehrenb.) Bunge		P	Jan, Feb, Sept and Oct	Wild spread
265. *Lantana rugosa* Thunb	Verbenaceae J.St.	N + P	Feb, Mar, Aug and Sept	Wild spread
266. *Zygophyllum bruguieri* (DC.) Christenh. & Byng (=*Fagonia bruguieri* DC.)	Zygophyllaceae R.Br.	N	Mar, May and June	Wild spread
267. *Zygophyllum indicum* (Burm.f.) Christenh. & Byng (=*Fagonia indica Burm*.f.)		N	Nov and Dec	Wild spread
268. *Zygophyllum simplex* L. (=*Tetraena simplex* (L.) Beier & Thulin)		N	Dec and Jan	Wild spread
269. *Tribulus parvispinus* C.Presl		N + P	Apr and May	Wild spread
270. *Tribulus terrestris* L.		N + P	May, June, Oct and Nov	Wild spread
271. *Tribulus macropterus* Boiss		N + P	Feb, Mar, Apr, May and June	Wild spread

**Table 3 plants-12-01402-t003:** Nectar and Pollen flow during different seasons in Al-Baha region, Southwestern Saudi Arabia.

	Source Categories	N		N + P		P + N		P		Pro		N + P + Pro		Total
Seasons (Months)		No.	%	No.	%	No.	%	No.	%	No.	%	No.	%	
Spring (III, IV, V)		54	31.8	51	30.0	18	10.6	41	24.1	4	2.4	2	1.2	170
Summer (VI, VII, VIII)		36	40.4	27	30.3	9	10.1	12	13.5	2	2.2	3	3.4	89
Autumn (IX, X, XI)		32	33.3	23	24.0	11	11.5	24	25.5	3	3.1	3	3.1	96
Winter (XII, I, II)		50	39.1	36	28.1	12	9.4	24	18.8	3	2.3	3	2.3	128

N (Nectar), P (Pollen), Pro (Propolis).

**Table 4 plants-12-01402-t004:** Contingency analysis of flowering honey bee plants by seasons in Al-Baha region, southwestern Saudi Arabia.

Seasons	Yes	No	Test		
	No. (%)	No. (%)	DF	X2-Value	*p*-Value
Autumn (Fall)					
Spring	96 (19.32)	172 (29.91)			
Summer	170 (34.21)	98 (17.04)			
Winter	85 (17.10)	183 (31.83)	3	74.59	0.0001

No.: number of species; DF: degree of freedom; Yes: those species flower in the particular season; No: those species do not flower in the particular season.

## Data Availability

All data generated or analyzed during this study are included in this published article and its Appendix A.

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
