# Peer review of "Plant Species as Potential Forage for Honey Bees in the Al-Baha Mountain Region in Southwestern Saudi Arabia"

_plants, 2023, doi:10.3390/plants12061402_

Round 1

Reviewer 1 Report

The article titled "Towards developing a checklist of plant species as potential  Honey bee forage inhabiting Albaha mountainous Region, Southwestern Saudi Arabia" raises important issues related to biodiversity vs. life of honey bees that have a direct impact on human well-being.

However, it must be improved before publication. Most of my comments I put on the pdf file. Generally, the section Material and methods needs some explanations. Also the section Results it must be improved. It is poorly written with very little details and without conclusions. Authors have a lot of research results but it looks like they do not know how to describe them and how to "extract" the quintessence.

In my opinion, the manuscript needs major revision and only after that could be published in Plants.

Author Response

Response to reviewers` comments

First of all thanks to the editor and the reviewer for their hard work and comments which make our manuscript better to be published in Plants

Reviewer 1

Comments in abstract section:

Comment: Line 20: ??? In the section Material and methods it is mentioned about flower morphology...

Response:

The abstract is typing error. The flower morphology in this sense is the shape and orientation of the flowers that determine the accessibility of the flowers to the bees. It can be easily seen by necked eyes during bees’ are visiting.  We correct the word in the abstract to morphology

Comment: In keywords do not repeat the words given in the title.

Response:

We rewrote the keywords to do not repeat the words given in the title in the keywords as follows:

Keywords: Honey production; Nectar; Pollen; Plant checklist; propolis

Comments in materials and methods section:

Comment: I recommend to add map as regular figure, not as supplementary meterial.

Response:

We nested the image to the text as Figure 4 and improved its resolution to 300 dpi

Comment: When the study was conducted? Each year or years?

Response:

The study was conducted for three consecutive years from Jan. 2019 to Dec. 2021. We clarified this point in the material and method section as follows:

The study concentrated on surveying the bee plant species distributed throughout the Region for three consecutive years from Jan. 2019 to Dec. 2021.

Comment: Line112-114: This part needs more explanation:

  1. What does it mean exactly "were identified on flower morphology", "bees's action during foraging hours" etc.?
  2. How did you evaluate bees's action during foraging hours? Did you count bees working on a unit of area in a specific unit of time eg. 1 minute?
  3. Did you take pollen loads and identified pollen grains under microscope?
  4. What do you mean beekeeper's experience?

Response:

  • This is the researchers’ observation during data recording on how the bees accessed the nectars, pollens, and propolis there by marking the plants weather they are important to bees or not. However, it doesn’t mean that plants were only identified by looking in to the flower morphology of the plants, rather it has been confirmed by direct observation of the researchers while bees were visiting and crosschecking it with the beekeepers.
  • As the title of this paper implies, it is to identify and document bee plants of the region. Hence in this case, it is only to identify that the plant is used by the bees or not. The researchers had been observing how the bees take the resources from the flowers. When the bees send their proboscis to the flowers, it implies that, that specific plant is a nectar source while it is a pollen source if the bees tried to uptake the pollen powder in their body and collect it in the pollen basket by using their legs. It is also clear when the bees collect the propolis and can be observed by the researchers. However, it was not necessary to count bees working on a unit of area in a specific unit of time
  • We didn’t take pollen loads and identified pollen grains under microscope. For the current study, the identification of the plants as nectar and pollen source was as stated above.
  • We mean that in our trips in identified the plants I was accompanying the old beekeepers in the region, as the Al-Baha region has long been famous for beekeeping and honey production. The old beekeepers have extensive experience that deserves attention. In most corner of the world, beekeepers are equipped with the local knowledge and experience which shall be regarded even in scientific studies

All these point clarified in the material and method section as follows:

The plant species valuable to bees were identified based on flower morphology (The flower morphology in this sense is the accessibility of the flowers to the bees. It can be easily seen by nacked eyes during bees are visiting), bees’ action during foraging hours on the plants' flowers, pollen in the corbiculae marked, and the old regional beekeepers' experience (observising how the bees take the resources from the flowers. When the bees send their probosis to the flowers, that implies that specisic plant is a nectar source while it is a pollen source if the bees tried to uptake the pollen powder in their body and collect it in the pollen bascket by using their legs. It is also clear when the bees collect the propolis and can be observed by the researchers [31]). After data collection we classified the bee plants were divided into different groups according to the type of collected forages.

Comment: Line 114: Please explain how did you observe and evaluate the forage?

Response:

We had been observing how the bees take the resources from the flowers. When the bees send their probosis to the flowers, that implies that specific plant is a nectar source while it is a pollen source if the bees tried to uptake the pollen powder in their body and collect it in the pollen basket by using their legs. It is also clear when the bees collect the propolis and can be observed by the researchers. We clarified this point in the manuscript as mentioned in the previous comment

Comment: Line 116: Please explain, what was the criterion of division? What is the difference between N+P and P +N group?

Response:

The criterion of division was the plants’ reward to attract insects for pollination. Most plants have both nectar and pollen rewards; however, their rewards are not nectars and pollens at equal levels; some are majorly important as nectar or pollen sources. In stating the importance of the plants as P+N and N+P is simply to imply the level of rewards to the bees. When that particular plant is labeled as P+N, the plant is majorly important as pollen source and nectar as secondary.  And it will be vice-versal when the labeling is stated as N+P; majorly nectar source and followed by pollen source. and we clarified this point as follows:

After data collection we classified the bee plants were divided into different groups according to the type of collected forages as follows: Nectar (N) source, pollen (P) source, and both pollen and nectar source plants. The latter group, was symbolled according to the main dominance source as follows: N+P (the main source of the forage was nectar beside some pollen was observed) or P+N (the main source of the forage was pollen beside some nectar was observed).

Comment: Line 117: What does it mean exactly?

This sentence was put to ensure the accuracy of the data as the bees activity is quite fast, researchers need to be careful in monitoring the bees activities while visiting plants. This monitoring had been followed in most contingent vising hours of a day and for some consecutive days during the flowering periods to confirm and make sure the specific importances of plants thereby to be identified and documented as bee  plants and for more clarification to this point we added supplementary file containing some images to the bees on the forage plants flower and cited this supplementary data in the manuscript as follows:

Careful observations were made of the bees’ actions while collecting resources for bee forage source plant categorization. The propolis (Po) source plants were determined by the observations made on bees’ visits to vegetative parts of plants collecting the plant secretions from leaves, stems, fruits, and branches (supplementary data: images showing bees on the forage plants).

Comment: Line 119: Just observation? Did you see bees collecting propolis and forming loads?

Response:

According to the scope of the study, frequent observations in fact during the active foraging hours for some consecutive days in the flowering periods while bees visit individual plants were the method. Actually, researchers were crosschecking with beekeepers to confirm their observation. This method is a common practice in undertaking similar bees - plants relationship studies and for more clarification we cited a reference reported the bees behavior during forage as mentioned before

Comments in result section:

Comment: Please add some graph connecting with plant's family numbers as well as their names.

Response:

We agree with you that graphs are more elaborative than tables, but the nature of the data is bulky and difficult to summarize it in the form of graph. Also, the main scope of this work is the plants important for bees forage, and so we presented the results as a table indicating the number of species and families in each category for the bee forage source, and Table 2 indicates clearly the detailed family -species classifications of the identified honeybee plants.  .

Comment: Table N % What does it mean this abbreviation?

Response:

This means the plant numbers and percent. We used No. for the number abbreviation instead of N to prevent the confusion with N as abbreviation for the nectar.

Comment: As was mentioned in the section Material and methods, what is the difference between these two groups? It should be necessary to study the amount of nectar and pollen produce by flowers if we would like to say that they are better or worse source of these products.

Response:

the difference between N+P and P +N group is in the dominant source of forages, in other words when the bee comes to the flower which part preferred as forage source and we clarified this point as follows:

After foraging period we examined the collected forage under microscope and classified the bee plants were divided into different groups according to the type of collected forages as follows: Nectar (N) source, pollen (P) source, and both pollen and nectar source plants. The latter group, was symbolled according to the main dominance source as follows: N+P (the main source of the forage was nectar beside some pollen was observed) or P+N (the main source of the forage was pollen beside some nectar was observed).

Comment: Should be given hear (and in other colums too), the information that first column it is number (No.) and the second one percentages (%).

Response:

We reformatted the tables as suggested

Comment: Line 148 add in Al-Baha region)

Response:

We added the region for more specification

Comment: Table 2: N% as above

Response:

This means the plant numbers and percent. We used No. for the number abbreviation instead of N to prevent the confusion with N as abbreviation for the nectar.

Comment: Table 2: Spring (3,4,5) It would be better to use Roman numerals for months

Response:

We used Roman numerals for months

Comment: Line 152: What was the reason to compare flowering and non-flowering plants while you are focuse on plants as bee sources?

Response:

In this case the flowering and non-flowering was compared the status of the plant with respect to the seasons. to avoid ambiguity "Non-flowering" shall be replaced by "not -flowering" as follows:

The variations in the distribution of flowering and not-flowering status of plant species

Comment: Please add the bloom calendar eg. as in following article: https://acpa.botany.pl/Plants-foraged-by-bees-for-honey-production-nin-northern-India-The-diverse-flora,118959,0,2.html.

Response:

We agree with you in this point that the bloom calendar in this article is clear and interesting but our study contains 271 plant species which make it will be missy if we added a bloom calendar in the suggested article which contain about 50 plant species. The calendar became more obvious when we replace the appendix with table 2 in the main text

Comment: In the tables 1 and 2, the abbreviation "N" meant nectar and in the table 3 the letter "N" is used for number. Please distinguish this.

Response:

We used No. for the number abbreviation instead of N to prevent the confusion with N as abbreviation for the nectar.

Comment: Figure 3: The word "endemism" it is not necessary here.

Response:

We deleted this word

Comment: Line 182: Please, say some words about species which were identified in your study. Which were the most? Etc.

When we replaced the appendix with Table 2 this information became more obvious and clear and for more clarification we added some information cited about species which were identified as follows:

Results of the current study showed that the wild plant species constitute 84.70%, followed by regional endemic (10.45%), near-endemic (2.99%), and endemic plants (1.87%) of the bee flora of the Region. Many bee forage plants like Ziziphus spina-christi, Vechilia species Senegalia asak, and Senegalia hamulosa are designated as rare and endangered. Some plant species are rare such as Blepharis edulis and Hypoestes forskaolii, and are considered valuable honey sources with high rates (Figure 2), (Table 2).   

Comments in Discussion and conclusion sections:

Comment: Line 209: replace by which with . They found that

Response:

We corrected this

Comment: Line 228: begine the sentence with In our study the spring...

Response:

We performed the suggested correction.

Comment: Line 232: Would be useful to add some meterological factors in the section Material and methods eg. average temperature, rainfall etc.

Response:

We agree with you in this point in the study for a year, but as the study was through different 3 years and take a long period it will be confusing and not add significant information.

Comment Lines 242-247: Some parts of the text from Conclusions should be linked to the section Results.

Response

We deleted these parts from the conclusion and add it to the result section as mention before in results comments

Comment: Instead of appendix would be better to add table which would be attached to the main text.

Response:

We replaced the appendix with Table 2 inside the main text.

Author Response

Response to reviewers` comments

First of all thanks to the editor and the reviewer for their hard work and comments which make our manuscript better to be published in Plants

Reviewer 2

Comment: Shorten the title. Suggest: Plant species as potential forage for honey bees in the Al-Baha mountain region in southwestern Saudi Arabia

Response:

It is a good suggestion we changed the title as suggested

Comment: Line 14: in abstract delete word background (line 14), methods (line 18), results (21), conclusion (25).

Response:

We corrected this as required

Comment: Line 28: write keywords in lower case

Response:

We write the keywords in lower case and changed some keywords to do not repeat the words given in the title in the keywords as follows:

Keywords: Honey production; Nectar; Pollen; Plant checklist; propolis

Comment: Line 60: Australia, Turkey, Mexico, Argentina…please insert the reference at the end of sentence

Response:

This sentence is linked to the previous section in which we talked about the honey bee production in Saudi arabia, so we transferred reference No. [2] to the end of theses sentence to clarify this point.

Comment: Line 100: include supplementary material in the main text, but improve the image quality

Response:

We nested the image to the text as Figure 4 and improve its resolution to 300 dpi

Comment: Line 125: excel into Excel

Response:

Thanks, we revised this typing mistake

Comment: Line 154: insert space between line 154 and Figure 1 and Figure 1 and line 155

Response:

Thanks, we revised this typing mistake

Comment: Line 204: Nectar into nectar

Response:

Thanks, we revised this typing mistake

Comment: Line 286: + space Pollen

Response:

Thanks, we revised this typing mistake in the table we made instead of the appendix

Comment: Please arrange the paper according to the Word template: 1. Introduction, 2. Results, 3. Discussion, 4. Materials and Methods, 5. Conclusions

Response:

We rearranged the paper as required

Comment: Technically refine the Tables according to the Template

Response:

We revised the tables according to the template

Comment: Also the reference list is not written according to the instructions.

Response:

We revised the references list according to the journal instructions

Response to reviewers` comments

First of all thanks to the editor and the reviewer for their hard work and comments which make our manuscript better to be published in Plants

Reviewer 2

Comment: Shorten the title. Suggest: Plant species as potential forage for honey bees in the Al-Baha mountain region in southwestern Saudi Arabia

Response:

It is a good suggestion we changed the title as suggested

Comment: Line 14: in abstract delete word background (line 14), methods (line 18), results (21), conclusion (25).

Response:

We corrected this as required

Comment: Line 28: write keywords in lower case

Response:

We write the keywords in lower case and changed some keywords to do not repeat the words given in the title in the keywords as follows:

Keywords: Honey production; Nectar; Pollen; Plant checklist; propolis

Comment: Line 60: Australia, Turkey, Mexico, Argentina…please insert the reference at the end of sentence

Response:

This sentence is linked to the previous section in which we talked about the honey bee production in Saudi arabia, so we transferred reference No. [2] to the end of theses sentence to clarify this point.

Comment: Line 100: include supplementary material in the main text, but improve the image quality

Response:

We nested the image to the text as Figure 4 and improve its resolution to 300 dpi

Comment: Line 125: excel into Excel

Response:

Thanks, we revised this typing mistake

Comment: Line 154: insert space between line 154 and Figure 1 and Figure 1 and line 155

Response:

Thanks, we revised this typing mistake

Comment: Line 204: Nectar into nectar

Response:

Thanks, we revised this typing mistake

Comment: Line 286: + space Pollen

Response:

Thanks, we revised this typing mistake in the table we made instead of the appendix

Comment: Please arrange the paper according to the Word template: 1. Introduction, 2. Results, 3. Discussion, 4. Materials and Methods, 5. Conclusions

Response:

We rearranged the paper as required

Comment: Technically refine the Tables according to the Template

Response:

We revised the tables according to the template

Comment: Also the reference list is not written according to the instructions.

Response:

We revised the references list according to the journal instructions

Round 2

Reviewer 1 Report

The Authors revised the paper in accordance with most of the comments given
previously.
There are still few small things to improve or change. I posted some
comments on pdf files.
I really like the supplementary file and in my opinion it will be useful for
readers.
I reccomend to publish the article after minor revision.

Author Response

Response to reviewers` comments

First of all thanks to the editor and the reviewer for their hard work and comments which make our manuscript better to be published in Plants

Comment: Line 88: Please, add a comment about plant families given in the Table 2.  Eg. Representatives of which  family were in the majority? What kind of  forage source do their flowers offer?

Response:

We added a comment about the major families and the kind of forage source their flowers offer to the result section as follows:

In the current study, about 62 families were recorded as a source of Nectar, pollen, and both nectar and pollen source plants.   Asteraceae has the highest number of individuals, 35 plant species, all of which are considered as sources of nectar, pollen or both.  The Fabaceae family followed the Asteraceae in the bee plant documentation in the region while still the Lamiaceae, Malvaceae,  and the Amaranthaceae found to contribute significantly in that order of importance in contributing honey bee resources in the form of nectar and pollen. Besides, Euphorbiaceae, Plantaginaceae, Boraginaceae, Brassicaceae and Acanthaceae were also among the top ten plant families holding major shares in the contribution of honey bee resources in the region. These ratings were based on the number of individual plant species contributing to the bee forage resource in the region. (Table 1 and Table 2).

Beside we added a discussion to these added comment to the discussion section as follows:

Furthermore, the outcomes of this study showed that about 268 plant species under 62 families were recorded in the study area. They are there for providing bee floral rewards, pollen, nectar, and propolis either in a combined or separate manner in that order of importance of contribution to the wellbeing of honey bees. Similar floral richness to the current study related to different geographic, edaphic, and environmental factors was also reported by Al-Aklabi et al. [24].  The fact is that the research location has varied geographical and climate elements that could explain the variance in the diversity of flowering plants throughout time. The most plant families observed in the region were the Asteraceae, Fabaceae, Lamiaceae and Malvaceae. These families found to have the highest number of species in Saudi Arabia probably due to their long-developed stress and drought tolerance abilities.  Similarly, Amaranthaceae, Euphorbiaceae, Plantaginaceae, Boraginaceae, Brassicaceae and Acanthaceae are known in their adaptation to arid and semi-arid regions; thus holding major shares in the honey bee resource contribution in the current region, which is mainly characterized by dry climatic situation. Similar findings were reported by Kuppler et al. [25] and Zurbuchen and Müller [26]

Comment: Table 2: All latin names should be in italics!!!

Response:

Sorry for this formatting error, we corrected all such typing errors in the whole manuscript.

Comment: Line 105: Pollen to pollen

Response:

Sorry for this typing error, we corrected all such typing errors in the whole manuscript.

Comment: Table 3 enter between III and IV

Response:

Sorry for this typing error, we corrected all such typing errors in the whole manuscript.

Comment: Line 216: Please, delete one dot.

Response:

Sorry for this typing error, we corrected all such typing errors in the whole manuscript.

Comment: Line 225-227: replace this part with availability to be “flower availability”

Response:

We replaced it to be flower availability

Comment: Line 128: “plants` flowers” delete plants`

Response:

We deleted this unnecessary word

Comment: Line 128 replace “pollen in the corbiculae marked” with “presence of pollen in the corbiculae”

Response:

We replaced this as required to be more obvious

Comment: line 230 do you mean specific?

Response:

Yes, this is typing error and we changed it

Comment: 233: delete “by using their legs” as this is obvious

Response:

We deleted this

Comment: Line 234: delete “were divided”

Response:

We deleted this unnecessary phrases

Comments in supplementary file: uniform the formatting of the scientific names all over the file and some typing errors:

Response:

We revised the scientific names typing and formatting all over the supplementary file.
